Effect of ecological restoration on carbon storage of damaged mountain slope in China’s East Ussuri River Basin

http://orcid.org/0000-0002-9925-737X Zhu Jianjun 1 2
Zhang Shilei 2
Chen Chen 2
Li Chunlin 2 cgglb@greensum.com.cn
1 State Key Laboratory of Microbial Technology, Shandong University , Qingdao , China
2 Greensum Ecology Co., Ltd. , QingDao , China
Phairuang Worradorn
Electronic publication date: 2025 Aug 8
Publication date: 2025
Volume: 13
Electronic Location ID: e19854
Received 2025 Jan 22; Accepted 2025 Jul 15
Copyright: © 2025 Zhu et al.
Copyright year: 2025
Copyright holder: Zhu et al.
License: This is an open access article distributed under the terms of the Creative Commons Attribution License, which permits unrestricted use, distribution, reproduction and adaptation in any medium and for any purpose provided that it is properly attributed. For attribution, the original author(s), title, publication source (PeerJ) and either DOI or URL of the article must be cited.
License URL: https://creativecommons.org/licenses/by/4.0/

Keywords: Ecological restoration, Carbon storage, Damaged slopes, Vegetation recovery, Soil reconstruction

Funding: Qingdao Postdoctoral Project QDBSH20240101048 Key Project of Qingdao Enterprise Technology Innovation 2320002000264 This work was supported by the Qingdao postdoctoral project “Characteristics and diversity of soil microbial communities in the ecological restoration process of damaged slopes” (Project Number: QDBSH20240101048), and the financial support and assistance from the key project of Qingdao enterprise technology innovation “Research on carbon sink estimation in aggregate spray seeding repair areas” (Project Number: 2320002000264). The funders had no role in study design, data collection and analysis, decision to publish, or preparation of the manuscript.

==============================
Ecological restoration techniques are extensively employed in the ecological restoration of damaged mountain ecosystems via effectively restoring the plant community and improving soil functions. Aggregate spray seeding technology as an effective ecological restoration method, can quickly restore the damaged slopes to their previous status and keep the ecosystem functions. However, the lack of understanding of the characteristic of carbon storage as well as its influencing factors limits the scientific management of carbon sink function of the rehabilitated damaged mountain ecosystem. In this study, relying on field surveys in the East Ussuri River Basin, we analyzed the carbon storage distribution and its influencing factors in damaged mountain slopes that had been restored through the spray seeding technology after an 8-year restoration phase. The results showed that the carbon storage distribution of the damaged slopes repaired by aggregate spray seeding is in the order of soil layer > tree layer > shrub layer > litter layer > herbaceous layer. Among them, the carbon storage of the vegetation layer is mainly distributed in the tree layer, and its size is similar to that of undisturbed natural slopes. Plants and soil are the main factors affecting the carbon storage of the repaired slopes, where the plant density has a significant positive correlation with carbon storage, accounting for 19% of the carbon storage variation, and the soil bulk density has a significant negative correlation with carbon storage, accounting for 23.7% of the carbon storage variation. This study reveals the mechanism of the aggregate spray seeding technology in restoring the carbon storage of damaged slopes and points out that regulating vegetation density and improving soil conditions are key to enhancing the carbon sink capacity of slopes.

Introduction

The Ussuri River Basin, situated in northern China, serves as a vital forestry region with abundant forest resources and significant forest coverage, playing a critical role in regional ecological carbon sequestration. Rapid urbanization and associated anthropogenic activities, particularly in mountainous areas, have led to extensive slope exposure and ecosystem degradation. According to the statistical yearbook data of Heilongjiang Province from 2001 to 2019, urban land increased from 10,418.25 km2 in 2001 to 10,973.75 km2, accompanying with forest area reduced from 1,070,843.75 km2 to 1,020,000 km2 (Jia & Yang, 2023). These changes have substantially compromised the region’s carbon sequestration capacity, Li et al. (2023) reported that a 2% reduction in forest land area in the Northeast black soil region between 2000 and 2020 resulted in a 1.65% decline in forest carbon absorption. Consistent with the China’s national commitment to peak greenhouse gas emissions by 2030 and achieve carbon neutrality by 2060 (Chen et al., 2022), enhancing the carbon sink of damaged mountain ecosystems represents a critical strategy for mitigating atmospheric CO2 increases and advancing China’s carbon neutrality goal. Therefore, it is of great significance to assess the characteristic of the rehabilitated ecosystem and quantitatively analyze the relevant influencing factors influencing their carbon sequestration potential.

Damaged mountain slopes, characterized by challenging site conditions such as shallow soils, water scarcity, and low fertility, present significant obstacles to ecological recovery (Hüttl & Frielinghaus, 1994; Xu et al., 2005; Ahirwal & Maiti, 2016; Egli & Poulenard, 2016; Moos et al., 2018; Hu et al., 2021; Wang et al., 2022; Shen et al., 2023). In response, extensive ecological restoration efforts have been implemented to reshape topography, rehabilitate soils, and reconstruct vegetation, ultimately aiming to restore the structure and function of these ecosystems (Aerts & Honnay, 2011; Prach, Jongepierová & Řehounková, 2013; Giupponi et al., 2019; Cai et al., 2019; Deivaseeno & Normaniza, 2021). Concurrently, various technologies have been developed and applied in the restoration of degraded slopes, including planting (Pawelek et al., 2015), hydraulic spray seeding technology (Albaladejo Montoro et al., 2000; García-Palacios et al., 2010), vegetation concrete (Liu et al., 2022), three-dimensional net (Zhong, Zhang & Wang, 2007; Zhang et al., 2017) and grass planting slope protection technology (Tai et al., 2024). Aggregate spray seeding (ASS)—a subset of hydraulic spray seeding technology—is a widely adopted method for restoring damaged mountain slopes (Shen et al., 2023), has demonstrated success in semi-arid and Mediterranean regions (García-Palacios et al., 2010; Cereno, Tan & Uy, 2011; Zhang et al., 2018; Chen et al., 2024), subtropic island (Wang et al., 2024a, 2024b). This technique rapidly promotes target plant growth, accelerates plant community succession, and ultimately achieves ecosystem restoration with self-sustaining capacity (Parsakhoo, Jajouzadeh & Motlagh, 2018; Emeka et al., 2021; Li, Wang & Xu, 2024).

Ecological restoration projects are one of the main drivers of China’s terrestrial ecosystem carbon sinks (Lu et al., 2018; Zhou et al., 2020; Ning et al., 2021). Comparing with other methods such as strengthening national land planning and controlling its use, implementing nature conservation (He et al., 2011, 2019; Yu et al., 2022), it proves an useful way to enhance China’s terrestrial ecosystem carbon sink (Feng et al., 2013; Lu et al., 2018; Shi, Chen & Feng, 2020; Zhou et al., 2020; Chen et al., 2025). Lu et al. (2019) found that ecological restoration projects contributed to 56% of the terrestrial carbon sink in the project implementation area.

The formation of ecosystem carbon sinks is a complex process involving plant carbon allocation and soil carbon sequestration. Vegetation, as a key factor driver in ecosystem carbon storage, gradually enhances its carbon sequestration capacity as plant communities establish and mature. Through photosynthesis, plants accumulate carbon while also transferring organic matter to the soil through litter and root exudates, thereby influencing soil organic carbon dynamics (Shi et al., 2015; Ahirwal & Maiti, 2018; Yang & Bai, 2021; Wu et al., 2023). Soil further acts as a critical carbon reservoir, where physical and chemical properties—such as nutrient availability, water retention, and erosion resistance—directly stabilize organic carbon and regulate long-term storage (Jia, Zhang & Yu, 2022). ASS combines seed banks of pioneer species with soil amendments like cellulose mulch to rapidly establish vegetation, fostering self-replacement of ecosystems, significantly altering the synergistic evolution process between vegetation and soil (Zhang et al., 2018; Jia, Zhang & Yu, 2022; Li, Wang & Xu, 2024), thereby shaping the unique characteristics of vegetation and soil carbon sinks. However, its application in monsoon-influenced temperate ecosystems—particularly for long-term carbon sink restoration—remains underexplored.

In the eastern Ussuri River Basin, a critical carbon sequestration zone in northeastern China (Sui et al., 2025), road construction has degraded mountain slopes into rocky, eroded landscapes, exacerbating carbon loss. The adaptation of ecological restoration to this region’s unique ecological context—monsoonal rainfall, dark-brown soils, and Ulmus-Betula secondary forests (Liang et al., 2022; Liu, Liao & Chang, 2024)—offers a tailored approach to underline specific and complex carbon storage properties. We hypothesize that aggregate spray seeding outperforms planted forests in restoring carbon storage to levels akin to undamaged slopes. This study evaluates an 8-year restoration project in the East Ussuri River Basin, analyzing carbon storage characteristics across vegetation layers (tree, shrub, herb) and soil profiles. By identifying its influencing factors in these restored slopes, we aim to (1) quantify ASS’s effectiveness in rebuilding carbon sinks and (2) provide actionable strategies for optimizing slope restoration in monsoon-vulnerable ecosystems. Our findings bridge a critical gap in mountain ecosystem research while advancing practical frameworks for carbon sink management under global climate goals.

Materials and Methods

The survey site

The survey site is located in the east of Ussuri River Basin (48°22′N, 134°28′E, 40–60 m altitude) (Fig. 1). The region is affected by monsoons, with cold and dry winters, and warm and humid summers. The annual average temperature is 3.4 °C and the annual average precipitation is 954.7 mm. The natural ecosystem consists of dark brown soil and a secondary forest dominated by Ulmus pumila and Betula platyphylla (Liang et al., 2022; Liu, Liao & Chang, 2024).

Figure 1 Location of site.

The survey site is located in the east of Ussuri River Basin (48°22′N, 134°28′E, 40–60 m altitude).

In the study region, roads construction has resulted in severely degraded mountain slope (Fig. 2). To address this environmental challenge, two restoration strategies were implemented in 2015: (1) Aggregate spray-seeding restoration technology (ASS) Developed by Qingdao Greensum Ecology Co., Ltd., China (Chen et al., 2024), this technique involved spraying a 7–10 cm growth medium composed of clay, organic additives, soil amendments, and necessary slow-release fertilizers. To facilitate natural succession, the seed bank incorporated native plants (Ulmus pumila, Betula platyphylla), and pioneer species (Lespedeza davurica, Caragana microphylla, Hippophae rhamnoides Linn, and Amorpha fruticosa). This multispecies approach targets both immediate erosion control (pioneers) and long-term carbon sequestration (native trees); (2) planted forests. This approach focused on manually planting native saplings (Ulmus pumila, Betula platyphylla) with pit planting technique (Jo & Park, 2017). Both restoration projects were supplied with regular irrigation and weeding during the first 3 years to ensure establishment.

Figure 2 The specific situation of different slope types.

(A) Unrepaired slope; (B) aggregate-sprayed restored slope; (C) planted forest slope; (D) undamaged natural slope.

Site design

Field surveys were conducted in August 2023 during the peak vegetation growth season. We selected three slope types: aggregate-sprayed restored slope (S1), planted forest slope (S2), and undamaged natural slope (S3). We established three 20 × 20 m2 survey plots in each slope type (nine plots total). Plant community assessments targeted the tree layer, shrub layer, herbaceous layer, dead wood layer, and litter layer. A nested quadrat design was employed to assess plant community: a central 10 × 10 m2 quadrat was surveyed tree layer for species composition, height, and diameter at breast height (DBH). A nested 2 × 2 m2 quadrat within the tree quadrat was used to record shrub composition and diameter at breast height (DBH). A 1 × 1 m2 subplot within the shrub quadrat quantified herbaceous herbaceous composition. Dead wood volume (decay stages) and litter thickness/biomass were measured within the 10 × 10 m2 quadrat. In our survey, trees and treelets were identified as DBH ≥2 cm, others were treated as shrub (Deb, Roy & Wahedunnabi, 2015).

Aboveground biomass samples (trees, shrubs, herbs) were collected, placed in labeled paper bags, and oven-dried at 65 °C for 48 h. Dried samples were ground (<0.2 mm) for carbon (C) concentration analysis. Given the rocky substrates and shallow soil profiles (≤10 cm depth), composite soil samples were collected from five systematically randomized cores (0–10 cm depth) per plot. Samples were homogenized, air-dried, sieved (2 mm), and analyzed for biochemical properties. Plant and soil C content was determined using via oxidation with potassium dichromate under acidic conditions, followed by titration measurement, soil N content was determined using the Kjeldahl method, and soil P content using the phosphorus molybdenum blue photometry method after digested in sulfuric acid-hydrogen peroxide (Nelson & Sommers, 1973).

Carbon storage calculation

The carbon storage of plant tree layer is calculated using Eqs. (1)–(6) (Eggleston et al., 2006; Zeng, 2017). The aboveground biomass of trees is calculated using Eq. (1) (Zeng & Tang, 2012). Use Eqs. (2)–(4) to derive Eq. (5) (Zeng & Tang, 2011), and the belowground biomass is calculated using Eq. (5).

(1) Ma=a×Db

where Ma is aboveground biomass (kg), a is parameter (a = 0.3ρ), ρ is basic wood density (g/cm3), D is diameter at breast height (cm), b is parameter (b = 7/3).

(2) Ma=a1×Da2+εMa

(3) Mb=b1×Db2×M^a+εMb

(4) R=b1×Db2+εR

where Ma is aboveground biomass (kg), Mb is belowground biomass (kg), R is the root to shoot ratio (the ratio of belowground to aboveground biomass), D is diameter at breast height (cm), M^a is the predicted value of aboveground biomass, ai and bi are model parameters, εi is the error term.

(5) M^b=c1×Dc2

where c1 = a1 × b1 = 0.0599, c2 = a2 + b2 = 2.16; M^b is the predicted value of belowground biomass.

(6) Ctree=(Ma+M^b)×CFtree

where Ctree is the carbon storage per unit area (kg/m2), Ma is aboveground biomass (kg), M^b is the predicted value of belowground biomass, CFtree is the organic carbon content (g/kg).

The carbon storage of shrub layer, herb layer, dead wood layer, and litter layer are calculated using Eq. (7) (Eggleston et al., 2006):

(7) Cx=Bx×CFx

where C is the carbon storage per unit area (kg/m2), B is the biomass (kg/m2), CF is the organic carbon content (g/kg), and x is shrub, herb, dead wood, and litter.

The soil carbon storage is calculated using Eq. (8) (Eggleston et al., 2006):

(8) Cs=ρb×CFs×0.1

where Cs is the soil carbon storage per unit area (kg/m2), ρb is the soil bulk density (g/cm3), and CFs is the soil organic carbon content (g/kg), focusing on a 0.1 m soil thickness in this survey.

Data statistics

The carbon storage of aggregate spray seeding restored slopes, planted forests restored slopes, and naturally undamaged slopes is represented by the average values of three sample plots. To analyze the relative contributions of plant density and soil physicochemical properties to ecosystem carbon storage, the R language relaimpo package is utilized. Differences in measured variables across slope types were assessed using one-way ANOVA with Tukey’s post-hoc test applied where ANOVA results were significant (P < 0.05). Linear regression analysis via the Origin analysis tool was performed to evaluate relationships between ecosystem carbon storage and soil bulk density, soil total nitrogen, and plant density, thereby elucidating key drivers of carbon sequestration. All statistical analyzes were performed in R version 3.6.1 (R Core Team, 2019).

Results

Current status of vegetation and soil carbon storage

The total carbon storage differed significantly (P < 0.05) across restoration types, with S3 (undamaged slope) exhibiting the highest storage (10.64 kg/m2), followed by aggregate spray seeding restored slope S1 (5.95 kg/m2) and planted forest slope (1.14 kg/m2) (Fig. 3, Table 1).

Figure 3 The total carbon content of different slope types.

The total carbon differed significantly (P < 0.05) across restoration types, with S3 (undamaged slope) exhibiting the highest storage (10.64 kg m−2), followed by aggregate spray-seeded slope S1 (5.95 kg m−2) and planted forest slope S2 (1.14 kg m−2). Different letters indicate significant differences (P < 0.05).

Table 1 Analysis of variance (ANOVA) results for carbon density across slope ecosystem components.

Parameters	P-value	
Tree layer	<0.001	
Shrub layer	0.348	
Herbaceous layer	0.022	
Litter layer	<0.001	
Root layer	<0.001	
Soil layer	<0.001	
Total carbon	<0.001	
Note:

P-values from one-way ANOVA comparing carbon density among aggregate-sprayed restored slope (S1), planted forest slope (S2), and undamaged natural slope (S3) ecosystems (n = 3). Significance level P = 0.05.

Carbon partitioning revealed distinct patterns: At S1, aboveground storage was dominated by trees (1.53 kg/m2) with minimal shrub (0.02 kg/m2), herbaceous (0.0033 kg/m2) and litter (0.0057 kg/m2) contributions, while belowground pools were soil-dominated (3.93 kg/m2 vs. roots: 0.47 kg/m2). S2 showed negligible aboveground storage (shrubs: 0.0073 kg/m2; herbs: 0.0017 kg/m2; litter: 0.0014 kg/m2) with soil accounting for virtually all carbon (1.13 kg/m2). In contrast, S3 displayed strong tree carbon accumulation (1.59 kg/m2) and exceptional belowground soil storage (8.58 kg/m2), dwarfing root contributions (0.44 kg/m2) (Table 2).

Table 2 Carbon density in different slope ecosystems.

Type	Aboveground carbon density (kg m−2)	Belowground carbon density
(kg m−2)	
Tree layer	Shrub layer	Herbaceous layer	Litter layer	Root	Soil	
Aggregate-sprayed (S1)	1.53a ± 0.03	0.02a ± 0.01	0.0033a ± 0.0002	0.0057b ± 0.0008	0.47a ± 0.03	3.93b ± 0.62	
Planted forest (S2)	N/A	0.0073a ± 0.001	0.0017b ± 0.0003	0.0014c ± 0.0004	0.0019b ± 0.0002	1.13c ± 0.009	
Undamaged natural (S3)	1.59a ± 0.001	0.013a ± 0.0006	0.0022b ± 0.0004	0.0138a ± 0.0004	0.44a ± 0.004	8.58a ± 0.008	
Note:

Values represent mean ± standard error (SE). N/A: Not Applicable (no tree layer present in S2). Different superscript letters within a column indicate significant differences (P < 0.05; one-way ANOVA with Tukey’s HSD post-hoc test).

Analysis of biological and soil physical and chemical properties in slope ecosystems

Vegetation density and structural attributes differed significantly among slope ecosystems (P < 0.05). The Shannon-Wiener index reached its maximum at S1, significantly exceeding values at S2 and S3 (which showed no statistical difference). Conversely, the Simpson index peaked at S2, followed sequentially by S1 and S3 (Fig. 4, Table 1). The total plant density varied markedly across sites: aggregate spray-seeded slope (S1) recorded 5.16 plants/m2, planted forest slope (S2) 2.35 plants/m2, and undamaged reference slope (S3) 3.47 plants/m2. At S1, tree, shrub, and herbaceous layer densities significantly surpassed those at S3, though tree diameter at breast height (DBH) was significantly lower than in the reference ecosystem (Tables 1, 3).

Figure 4 The Shannon-Wiener index (A) and the Simpson index of different repair types (B).

The Shannon-Wiener index reached its maximum at S1 (1.54), significantly exceeding values at S2 (0.64) and S3 (0.78). The Simpson index peaked at S2 (0.89), followed sequentially by S1 (0.73) and S3 (0.51). Different letters indicate significant differences (P < 0.05).

Table 3 Plant community characteristics in different slope ecosystems.

Type	Tree layer	Shrub layer	Herbaceous layer	
Density (plants m−2)	DBH (cm)	Density (plants m−2)	Density (plants m−2)	
Aggregate-sprayed (S1)	0.50a ± 0.03	5.49b ± 0.20	0.36a ± 0.02	4.30a ± 0.38	
Planted forest (S2)	N/A	N/A	0.08c ± 0.01	2.27b ± 0.35	
Undamaged natural (S3)	0.05b ± 0.002	15.00a ± 0.12	0.15b ± 0.01	3.27b ± 0.12	
Note:

Values represent mean ± standard error (SE). DBH: Diameter at Breast Height. N/A: Not Applicable (no tree layer present in S2). Different superscript letters within a column indicate significant differences (P < 0.05; one-way ANOVA with Tukey’s HSD post-hoc test).

Comparative soil analysis revealed distinct physicochemical gradients (Tables 1, 4). The aggregate spray-seeded slope (S1) exhibited significantly higher soil water content (SWC), total nitrogen (TN), total phosphorus (TP), and total potassium (TK) than both planted (S2) and undamaged (S3) slopes. Conversely, soil bulk density at S1 was significantly lower than at S2 and S3.

Table 4 Soil physical and chemical properties in different slope ecosystems.

Type	Bulk density
(g/cm-3)	Soil water content (V/V%)	Total nitrogen
(g kg−1)	Total phosphorus
(g kg−1)	Total potassium
(g kg−1)	
Aggregate-sprayed (S1)	0.54b ± 0.02	34.36a ± 0.89	1.64a ± 0.13	0.89a ± 0.08	18.68a ± 0.59	
Planted forest (S2)	0.75a ± 0.03	28.94b ± 0.41	0.50c ± 0.05	0.55b ± 0.04	14.12c ± 0.38	
Undamaged natural (S3)	0.71a ± 0.01	29.69b ± 1.31	0.91b ± 0.07	0.59b ± 0.01	15.78b ± 0.05	
Note:

Values represent mean ± standard error (SE). Different superscript letters within a column indicate significant differences (P < 0.05; one-way ANOVA with Tukey’s HSD post-hoc test).

Analysis of influencing factors on ecosystem carbon storage

Variance partitioning analysis revealed that plant and soil factors collectively explained 99.8% of carbon storage variation in aggregate-sprayed restored slope (S1), with vegetation contributing 19% to this model (Fig. 5). Soil physicochemical properties dominated the explanatory power (80.8%), where soil bulk density emerged as the primary predictor (23.7%), followed sequentially by soil total nitrogen (19.2%), soil total potassium (14.8%), soil total phosphorus (11.7%), and soil water content (11.4%). The correlation analysis further demonstrated significant associations: ecosystem carbon storage showed strong negative correlation with soil bulk density (P < 0.01), while exhibiting a significant positive correlation with soil total nitrogen (P < 0.01) and plant density (P < 0.001) (Fig. 6).

Figure 5 Relative importance of factors in the changes of ecosystem carbon storage.

Plant and soil factors together account for 99.8% of the variation in carbon storage on the aggregate spray seeding restored slope. Soil bulk density (BD) explaining 23.7%, soil water content (SWC) explaining 11.4%, soil total nitrogen (TN) explaining 19.2%, soil total phosphorus (TP) explaining 11.7%, soil total potassium (TK) explaining 14.8% and plant density explaining 19%.

Figure 6 Influence of soil bulk density (A), total soil N (B) and plant density (C) on ecosystem carbon storage.

Ecosystem carbon storage (C) has a significant negative correlation with soil bulk density (BD) and a significant positive correlation with soil total nitrogen (TN) and plant density.

Discussion

Impact of ecological restoration on slope carbon storage

This study quantifies carbon storage patterns in damaged slopes of the Ussuri River Basin after 8 years of restoration. Our results reveal that ASS successfully reestablishes a carbon distribution profile analogous to undamaged reference slopes (Table 2), confirming its efficacy in restoring ecosystem carbon functions. The technique’s “tree, shrub, grass” plant configuration used in ASS can rapidly establish a stable plant community, (>5% vegetation coverage (Li et al., 2011)), with vegetation carbon dominated by trees (98%) consistent with natural forest ecosystems (Wang et al., 2011). Notably, tree layer carbon in ASS restored slopes matched undamaged slopes (1.53 vs. 1.59 kg/m2, Table 2), confirming recovery of vegetative sequestration function. However, this remains about 50% below Heilongjiang’s forest average 3.3 kg/m2 (Liu, Wang & Luan, 2011), attributable to the younger age and smaller DBH (Ren & Xia, 2017). Carbon accumulation is projected to increase with restoration duration (Shi et al., 2015; Ahirwal & Maiti, 2018), indicating significant future potential in plant carbon sequestration but require long-term monitoring.

This study found that the soil carbon storage of ASS restored slopes is 3.5 times that of planted forest restored slopes, confirming the effectiveness of ASS ecological restoration in increasing soil carbon storage (Xu, Qu & Wang, 2020). The reasons for this are related to the ASS restoration technology: firstly, the technology uses a substrate rich in organic matter to construct an artificial soil layer (Li et al., 2011), significantly increasing the organic carbon content in the soil; secondly, by establishing a high-density seed bank, ASS promotes the input of organic matter from seed residues, plant litter, and roots into the soil, further enhancing soil carbon storage (Wu et al., 2023); finally, soil erosion is a major factor in the loss of soil organic carbon (Liu et al., 2023), and the artificial soil structure constructed by ASS is stable and has strong erosion resistance (Li et al., 2011), effectively promoting the fixation of soil carbon. However, compared with undamaged slopes, the soil carbon storage of ASS restored slopes is still lower. This may be related to the low amount of forest litter return. The results of this study show that the litter carbon storage in the sprayed restoration area is only 41% of the carbon storage of undamaged slopes, consistent with the results of a study where natural forest land was converted to artificial forest (Ran, Xu & Wan, 2022). This indicates that the reduction in the amount of litter is a key factor leading to low soil organic carbon storage. In addition, another reason for the low soil carbon storage in aggregate spray seeding areas may be related to the thin artificial soil layer in the damaged slope area. Relevant studies have pointed out that in rocky desertification ecosystems, due to high rock exposure rates and thin soil, the soil carbon storage after restoration is far below the national average (Lu et al., 2019). A study by Lei et al. (2023) also found that the soil organic carbon pool in the spoil dump slopes of northern mining areas still has a lower content than the soil in natural areas after restoration.

Impact of ecological restoration on plants and soil

Vegetation restoration constitutes the primary objective in rehabilitating damaged slopes. This study revealed significantly higher plant density and Shannon-Wiener index at aggregate-spray restored slopes (S1) compared to planted forest restored slopes (S2) after 8 years and undamaged slopes (S3) (Table 3), indicating superior vegetation recovery of ASS method (Li et al., 2011). Soil water content and nutrient availability serve as key drivers of plant community composition, profoundly influencing species richness and diversity (Fehmi & Kong, 2012; Wu et al., 2014). Consistent with Bennie et al. (2008), we observed significantly elevated soil water content (SWC) and nutrient concentrations (TN, TP, TK) at S1 than that of the S2 and S3. These enhanced soil conditions likely underpin the observed differences in vegetation structure. The high-density seeding methodology intrinsic to aggregate spray seeding further contributed to these outcomes (Tamura et al., 2017). This approach facilitated rapid establishment of a multi-strata vegetation structure (trees, shrubs, and grasses) on bare slopes within 2 years. Plant density subsequently increased progressively during community succession from pioneer to late-stage species (Li et al., 2011).

Damaged mountain slopes, characterized by challenging site conditions such as shallow soils, water scarcity, and low fertility, present significant obstacles to ecological recovery (Hüttl & Frielinghaus, 1994; Xu et al., 2005; Egli & Poulenard, 2016; Moos et al., 2018; Hu et al., 2021; Wang et al., 2022; Shen et al., 2023), necessitates prioritizing soil quality improvement. Ecological restoration demonstrably counters soil degradation, enhances nutrient cycling, and sustains soil quality to facilitate ecosystem recovery (Wang et al., 2014; Li et al., 2016). Consistent with Morrison & Foster (2001), our findings indicate significantly higher levels of soil total nitrogen (TN), total phosphorus (TP), and total potassium (TK) at site S1 compared to sites S2 and S3. This nutrient enrichment primarily stems from enhanced litter inputs and root system dynamics. Critically, the deliberate incorporation of leguminous species at S1—noted for nitrogen-rich litter and roots—promoted soil aggregate formation and stability (Kumar et al., 2020). Enhanced aggregate stability significantly improves key soil physicochemical properties (Regelink et al., 2015), including reduced bulk density and increased water retention capacity (see Table 4). These improved soil physical properties subsequently enhance the stability of carbon and nitrogen pools (Ma et al., 2022). Coupled with rapid nutrient cycling rates characteristic of broad-leaved species, this significantly contributes to soil fertility enhancement (Shao et al., 2020). Furthermore, the artificial soil matrix employed in aggregate spray seeding technology, typically amended with organic matter, soil conditioners, and necessary slow-release fertilizers, provides an initial plant growth foundation. Subsequent vegetation recovery, driven by continuous accumulation of litter, root network development, and root exudation, further facilitates progressive improvement in soil physicochemical properties (Yang et al., 2022; Dan, 2023).

Analysis of influencing factors on ecosystem carbon storage

Ecological restoration effectively enhances ecosystem carbon storage through dual pathways: accelerated vegetation growth and promoted soil carbon sequestration (Li, Niu & Luo, 2012; Lange et al., 2015; Zheng et al., 2024). Plant density is an important variable influencing the forestry biomass accumulation (Marquard et al., 2009) by regulating resource competition and crown development (Postma et al., 2021). Contrary to self-thinning theory, which typically predicts negative density-biomass relationships due to competitive exclusion (Yoda, 1963; Ouyang et al., 2019; Hao et al., 2022), our study revealed a positive correlation between plant density and carbon storage (P < 0.001), explain 19% of the variation after 8-year restoration (Figs. 5 and 6). This apparent contradiction resolves when considering stand density context: at low densities (0.5 plants/m2), wider spacing reduces competition, enabling greater branch expansion and individual tree growth (Fanin et al., 2025). This maximizes per-unit biomass in late-successional stages. In addition, a relative high initial densities (e.t., via hydroseeding) enhances resource acquisition efficiency per unit area by promoting rapid canopy closure to capture more solar radiation, improving interception of water and nutrients before loss, suppressing weeds, and accelerating early biomass accumulation (Westgate et al., 1997; Tamura et al., 2017).

Our study found that carbon storage in aggregate-spray restored slopes is significantly negatively correlated with soil bulk density (P < 0.01), indicating that reducing soil bulk density promotes carbon accumulation on such slopes. This relationship arises because bulk density directly influences key soil properties governing carbon dynamics, including aeration, infiltration, water-holding capacity, and solute transport (Chai & He, 2016; Xu, He & Yu, 2016). Elevated bulk density can impede root development and plant growth, thereby reducing photosynthetic rates and aboveground productivity (Li & Wang, 2000; Liu & Shan, 2003). Consistent with this mechanism, site S1 exhibited a 24% lower bulk density than site S3 (Table 4), corresponding to more favorable vegetation conditions. This finding aligns with Chen et al. (2024), who demonstrated that aggregate-spray stabilization (ASS) effectively restructures soil on mining tailings ponds to achieve lower bulk density.

Conclusion

This study demonstrate aggregate spray seeding (ASS) restoration technology can restore carbon storage distribution pattern (soil > tree > shrub > litter > herbaceous) in damaged East Ussuri River Basin slopes over an 8-year period. Notably, carbon sequestration within the tree layer approached levels observed in undamaged natural slopes. We identified plant density (explaining 19.0% of variation, positive effect) and soil bulk density (explaining 23.7% of variation, negative effect) as the principal factors governing carbon accumulation on these restored slopes. This synergy between optimal plant spacing and soil amelioration creates conditions for maximal carbon capture in restored slopes. Future studies should further focus on the impact of litter on soil carbon storage during the process of vegetation restoration, and in conjunction with the global carbon neutrality goal, promote the restoration and enhancement of carbon sequestration functions in damaged ecosystems.

Supplemental Information

Supplemental Information 1 Raw data.

Additional Information and Declarations

Competing Interests

Jianjun Zhu, Shilei Zhang, Chen Chen, and Chunlin Li are all employees of Greensum Ecology Co., Ltd.

Author Contributions

Jianjun Zhu conceived and designed the experiments, performed the experiments, analyzed the data, authored or reviewed drafts of the article, and approved the final draft.

Shilei Zhang conceived and designed the experiments, authored or reviewed drafts of the article, and approved the final draft.

Chen Chen analyzed the data, prepared figures and/or tables, authored or reviewed drafts of the article, and approved the final draft.

Chunlin Li conceived and designed the experiments, authored or reviewed drafts of the article, and approved the final draft.

Data Availability

The following information was supplied regarding data availability:

Raw data, including soil data, carbon storage data, and plant community data, is available as a Supplemental File.

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
