# Peer review of "Effect of ecological restoration on carbon storage of damaged mountain slope in China’s East Ussuri River Basin"

_PeerJ, doi:10.7717/peerj.19854_

## Round 0.1 · original submission · Major Revisions

The data display (in table and graphical form) is still very basic.
Maps and photos need to be enhanced and highlighted with the research site.
The results are still explained by the discussion description that hasn't been enhanced by relevant literature.

**Language Note:** The review process has identified that the English language must be improved. PeerJ can provide language editing services - please contact us at [email protected] for pricing (be sure to provide your manuscript number and title). Alternatively, you should make your own arrangements to improve the language quality and provide details in your response letter. – PeerJ Staff

·

Basic reporting

There are some sentences that are unclear and ambiguous.
The article still needs to be enriched with literature references.
The presentation of data is still too simple (graphical form and table presentation).
Images and maps must be enriched and emphasized with the research location.
The presentation of the discussion description still explains the results that have not been enriched with supporting literature.

Experimental design

The method needs to be described in more detail, such as plot description, number of plots, test parameters, detailed plot locations, plot illustrations (photos).

Validity of the findings

Novelty and gap analysis are not clearly described in the introduction.
Hypothesis is not clearly described in the introduction.
The description of the conclusion needs to be improved.

Additional comments

-

·

Basic reporting

The manuscript entitled “Effect of Ecological Restoration on Carbon Storage of Damaged Mountain Slope in China’s East Ussuri River Basin” addresses a crucial issue in ecological restoration related to carbon storage, particularly in mountain ecosystems. The authors evaluated the results of the application of aggregate spray seeding technology as a restoration method after an eight-year period, providing a medium-term perspective that can enhance understanding of ecological dynamics. To my view, the study reinforces previous knowledge on how restoration techniques can influence carbon storage capabilities, aligning with current global climate change concerns. However, I see some important limitations for its publication in Peer J journal.
My main concern is about the lack of novelty of the study. The authors introduce in lines 81-84 the aggregate spray seeding technique as an innovative approach by citing Li, 2024. However, this technique has been previously tested and reported in degraded slopes worldwide (see for instance Garcia-Palacios et al., 2000 in Ecol. Engin., Albaladejo Montoro et al., 2000 in Land Deg. Develop., or Emeka et al., 2021 in Land, among many others).

Experimental design

I miss a significant lack of information regarding experimental design, methodology, and data analysis. For example, lines 121-122 discuss two restoration techniques, but only go into detail explaining one of them. The methodology followed in field sampling for soil and vegetation is not detailed. Nor is there any information provided about which variables were considered, other than those included in the carbon storage calculation formula. This section should provide all the information necessary to replicate the experiment, but this would not be possible in the current format. Similarly, the statistical analyses and models used to compare the results obtained on undegraded, naturally recovered, and restored slopes could be explained in greater detail.
The study lacks a starting hypothesis, however, the expected results regarding carbon stock distribution are fairly obvious. The authors highlight the role of soil and vegetation in carbon storage (lines 85-101 and Figure 2), but with all due respect, I don't understand what's new or surprising about this. Biomass and therefore the different structural levels of vegetation are expected to significantly impact carbon stocks.

Validity of the findings

The authors present detailed findings regarding carbon storage distribution across different layers (soil, tree, shrub, etc.), However, the results of the comparison between restored, naturally recovered and non-degraded slopes, which are in my opinion the strong point of the study, are simply shown in a table, which is not very visually attractive, when, in my opinion, it should be the central part of the article. Furthermore, if authors want to highlight the suitability of implementing the aggregate spray seeding technique to properly restore degraded slopes further comparative analyses with previously studied restoration methods could provide a more comprehensive understanding of the effectiveness of the technique applied.

Finally, regarding discussion, although the study reveals important data on carbon storage, it lacks a thorough exploration of the novelty of findings compared to existing literature. A more explicit comparison with previous studies would strengthen its claims of originality.

---

## Round 0.2 · accepted · Accept

This revised version is suitable for publication.